# Vulnerability of maize, millet, and rice yields to growing season precipitation and socio-economic proxies in Cameroon

Terence Epule Epule[1,2]*, Abdelghani Chehbouni[1,3], Driss Dhiba[1], Daniel Etongo[4,5], Fatima Driouech[1], Youssef Brouziyne[1], Changhui Peng[6,7]

1 International Water Research Institute (IWRI), Mohammed VI Polytechnic University (UM6P), Benguerir, Morocco, 2 Department of Geography, McGill University, Montreal, Quebec, Canada, 3 Centre for Remote Sensing Application (CRSA), Mohammed VI Polytechnic University (UM6P), Benguerir, Morocco, 4 James Michel Blue Economy Research Institute, University of Seychelles, Anse Royale, Seychelles, 5 Department of Environmental Sciences, University of Seychelles, Anse Royale, Seychelles, 6 Department of Biological Sciences, University of Quebec at Montreal (UQAM), Montreal, Quebec, Canada, 7 Centre for Ecological Forecasting and Global Change, College of Forestry, North East A & F University, Yangling, Shaanxi, China

* epule.terence@um6p.ma, terence.epule@mail.mcgill.ca

**Data Availability Statement:** The data presented in this study are available in the supplementary material section.

## Abstract

In sub-Saharan Africa growing season precipitation is affected by climate change. Due to this, in Cameroon, it is uncertain how some crops are vulnerable to growing season precipitation. Here, an assessment of the vulnerability of maize, millet, and rice to growing season precipitation is carried out at a national scale and validated at four sub-national scales/sites. The data collected were historical yield, precipitation, and adaptive capacity data for the period 1961–2019 for the national scale analysis and 1991–2016 for the sub-national scale analysis. The crop yield data were collected for maize, millet, and rice from FAOSTAT and the global yield gap atlas to assess the sensitivity both nationally and sub-nationally. Historical data on mean crop growing season and mean annul precipitation were collected from a collaborative database of UNDP/Oxford University and the climate portal of the World Bank to assess the exposure both nationally and sub-nationally. To assess adaptive capacity, literacy, and poverty rate proxies for both the national and regional scales were collected from KNOEMA and the African Development Bank. These data were analyzed using a vulnerability index that is based on sensitivity, exposure, and adaptive capacity. The national scale results show that millet has the lowest vulnerability index while rice has the highest. An inverse relationship between vulnerability and adaptive capacity is observed. Rice has the lowest adaptive capacity and the highest vulnerability index. Sub-nationally, this work has shown that northern maize is the most vulnerable crop followed by western highland rice. This work underscores the fact that at different scales, crops are differentially vulnerable due to variations in precipitation, temperature, soils, access to farm inputs, exposure to crop pest and variations in literacy and poverty rates. Therefore, caution should be taken when transitioning from one scale to another to avoid generalization. Despite these differences, in the sub-national scale, western highland rice is observed as the second most vulnerable crop, an observation similar to the national scale observation.

**Funding:** This work was supported by a research grant from Mohamed VI Polytechnic University to T.E under the auspices of the research grant to new professors under grant number UM6P. The link to the funder's website is: https://www.um6p.ma/en. The funder had no role in study design, data collection and analysis, decision to publish, or preparation of the manuscript.

**Competing interests:** The authors have declared that no competing interests exist.

# 1. Introduction

The Fifth Assessment Report (AR5) of the Intergovernmental Panel on Climate Change (IPCC) projected temperature increase in sub-Saharan Africa (SSA) of between 1.5–2.5˚ C up to 2050 based on the RCP 4.5 scenario [1]. In the last three and a half decades SSA witnessed a 0.2–2.0˚ C increase in annual surface air temperatures [2]. The implications of the above changes are parallel changes in growing season precipitation reflected mainly in unreliable and insufficient precipitation. This has occasioned climate changes in SSA and an increase in the vulnerability of current agricultural systems and worrying future projections of the latter [2–6]. In Cameroon, agriculture is essentially rainfed and changes in precipitation do affect the cultivation of crops [2–5, 37, 38]. Due to persistent droughts, the mean crop growing season precipitation in Cameroon is around 200 mm for maize, rice, and millet [38]. However, across Cameroon, there is a spatial variation in precipitation. For example, the south western part of the country records about 3000 mm of precipitation annually, the south east records about 1600 mm annually, the western highlands records 2000 mm annually and northwards it drops to about 500 mm annually [5, 6, 37]. As can be seen above, the annual precipitation exceeds the growing season precipitation due to persistent droughts. For example, in the northern Sahelian parts of the country, these droughts are recurrent and occur on a yearly basis thus placing a lot of stress on crops [5, 6, 37]. Even though droughts are recurrent in Cameroon, the most recent and significant droughts were recorded between 2012–2015 in which several tons of maize, beans, millet, rice seeds and seedlings were destroyed [3–6, 37,38].

Such challenges are even more daunting as some studies such as Giannini et al. [6] argue that the current dilemma of changing growing season precipitation is made even more difficult as the drivers of the vulnerability of food systems in most African countries including Cameroon go beyond climatic variables and include several non-climatic variables such as soils, slopes, crop pests, fertilizers inter alia [6, 7]. In this context and to better understand and estimate the level of vulnerability of a crop; it is important to use, test and develop novel approaches that reflect the role of climatic and non-climatic variables [7–9]. These dynamics are disturbing as they are already having profound effects on agriculture which is the "life wire" of most SSA economies including Cameroon [1, 3, 4].

The importance of assessing the vulnerability of crops through an index that heralds the role of climatic and non-climatic variables is already gaining grounds in academic scholarship. For example, Epule et al. [5] developed a crop yield vulnerability index for maize yields to droughts in Uganda; this approach was later used by Epule and New [10], to assess the vulnerability of five crops in 2019 in Uganda. The latter index is currently being tested in this current study in Cameroon. There are other indices that have been developed as follows: The Global Notre-Dame (ND-Gain) adaptation index [11], the crop-drought indicator [12], the water-poverty index which shows vulnerability to access water based on poverty [13] and the farmer vulnerability to global change index which focuses on large scale irrigation and maladaptation [14] (Table 1).

The index used in this study is mostly applicable in assessing vulnerability of cropping systems to droughts in an African crop production context by incorporating crop yield, climatic and adaptive capacity data. Its effectiveness has been tested mostly on maize, beans, cassava, groundnuts, potatoes, millet, and sweet potatoes in Uganda [5, 10]. This approach is advantageous as it can respond to the needs of providing a holistic approach and integrating climatic and socio-economic variables into vulnerability mapping and assessments. In addition, it uses historical data which reflects ground truthing and the actual stress that the cropping systems might be subjected to [15]. The adaptive capacity part of this index focuses on the use of two

**Table 1. Comparison of vulnerability indices.**

| Indices | Characteristics/Citations |
|---|---|
| The Global Notre-Dame Adaptation Index-ND Gain | Evaluates readiness for adaptation [11] |
|  | Leverages public and private sector investments [11] |
|  | Considers the following sectors: Health, Food, Ecosystem, Habitat, Water, and Infrastructure [11] |
|  | Frames vulnerability as a function of exposure, sensitivity, and adaptive capacity [11] |
| The Crop-Drought Indicator | Uses indicators of vulnerability such as precipitation and temperature [12] |
| Water Poverty Index | Bases vulnerability on poverty [13] |
|  | Considers water resource stress as an additional component of vulnerability [13] |
| Farmer Vulnerability to Global Change Index | Focuses on agricultural adaptations that result in maladaptation [14] |
|  | Irrigation projects as agents of maladaptation [14] |
| Vulnerability Index of Crop Yields to droughts** | Vulnerability is framed as a function of exposure, sensitivity, and adaptive capacity [5, 10] |
|  | Focus on the interactions between crops, climate, and socio-economic drivers [5, 10] |
|  | Adaptive capacity framed based on African realities [5, 10] |

**used in this current study Source: Authors' conceptualization.

proxies of adaptive capacity that are most relevant in the cropping systems equation of the country; these are literacy and poverty rates [16].

Based on the above, this current study aims at testing the vulnerability index by simulating the relative vulnerability of maize, millet, and rice at a national scale in Cameroon to growing season precipitation and socio-economic proxies like literacy and poverty rates. In addition, this work further validates the national scale perspective at four sites in Cameroon based on the availability of data. In other words, this study's objective is to assess the vulnerability of maize, millet, and rice in Cameroon by assessing the sensitivity, exposure, and adaptive capacity of these crops based essentially on growing season precipitation and two socio-economic proxies. The goal is therefore to use the findings to inform and support policy towards revamping the most vulnerable of these three crops and in providing a better understanding of the dynamics herein. As a country in SSA, Cameroon is both a breadbasket and a country which is invariably affected by climate change [17]. The country exibits huge changes in annual precipitation across its agroecological zones. For example, annual precipitation varies from about 3000 mm in the south western Cameroon plateau to close to 500 mm in the extreme north of the country (Fig 2). Such variations are likely to be equally reflected in the growing season precipitation of many crops which are often impacted by recurrent droughts [1].

As such, in this study, the above objectives and the selected crops are rationalized by the fact that 1. maize, millet, and rice are among the most widely cultivated and affordable crops in Cameroon [17, 18]; 2. these crops are among the most consumed staple foods in Cameroon [18]; 3. the cultivation of these crops is essentially in the hands of small-scale peasant farmers who are inadequately equipped to adapt to changes caused by climate [19, 20] and lastly, 4. huge gaps still exist in the context of analysing the vulnerability of these crops to droughts in Cameroon. Apart from a study by [18] that addresses the role of climatic and non-climatic drivers of maize yields in Cameroon, to the best of our knowledge there are no other studies

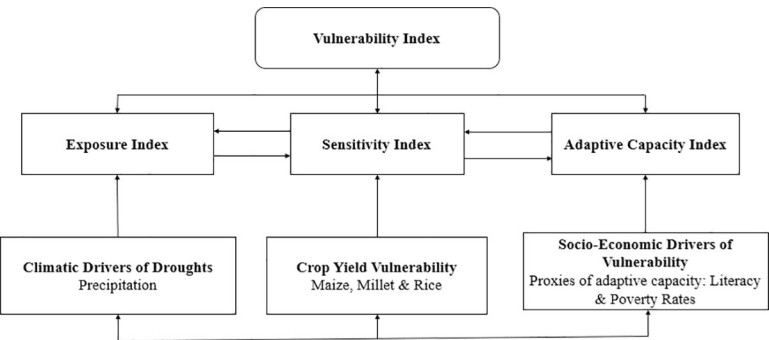

**Fig 1. Schematic representation of the vulnerability conceptual framework.** Source: Authors' conceptualization.

that have used this vulnerability index to assess the relative vulnerability of these three crops in Cameroon. Aside from the Ugandan case study, this approach has never been used before. However, other studies that have used similar approaches include those by Mishra and Singh [21, 22], who focus on precipitation deficits and temperature changes, Kamali et al. [23, 24]

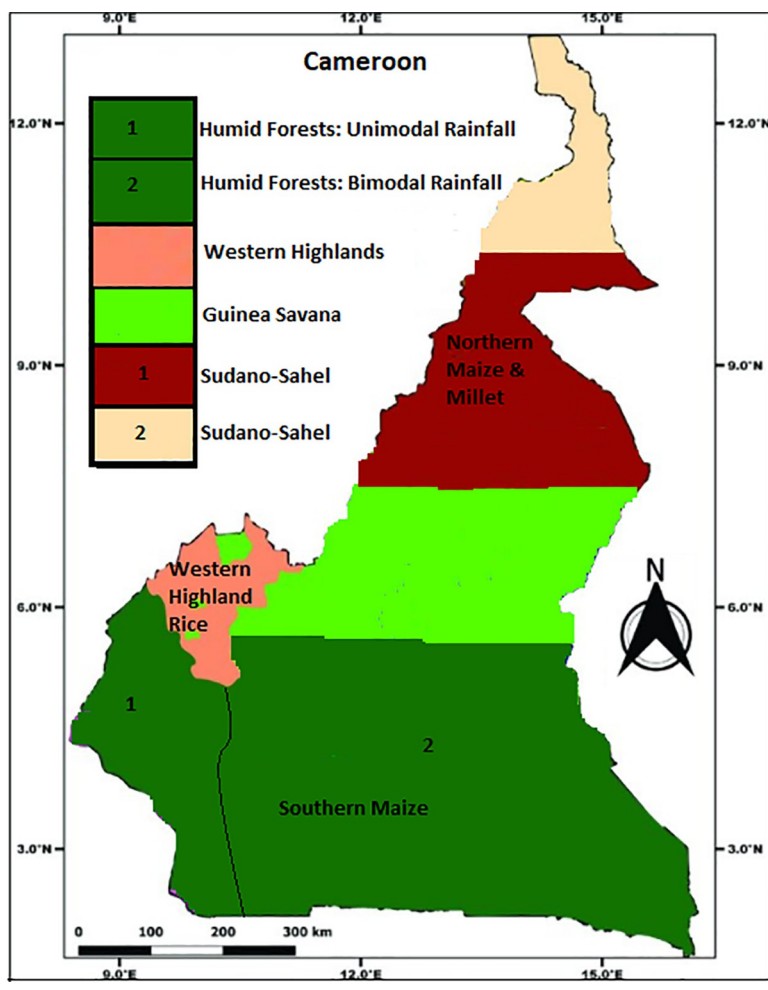

**Fig 2. Map of Cameroon.** Source: Authors' conceptualization.

who used social and physical perspectives to evaluate the vulnerability of maize yields to droughts in Africa, and Jassogne et al. [25] and Mwaura and Okoboi [26] who also used crop-climate interactions. The section that follows discusses the conceptual dimensions of vulnerability and the methods used in this study.

## 2. Conceptual framework

### 2.1. Vulnerability index

Vulnerability is the extent to which a system is susceptible and unable to cope with the negative effects of climate change and extreme weather episodes [2, 27] (Fig 1). In the context of crop yields, vulnerability is the extent to which crops cope with multiple stressors and shocks that are climatically and socio-economically driven. Based on the following studies [1, 3, 12, 27, 28], vulnerability is driven by: 1) the level of *exposure* [28, 29]; 2) the *sensitivity* [28, 29], and 3) the *adaptive capacity*, which is determined by the ability of the farmers involved to absorb the water stress caused by variations in precipitation [29–32]. Previously, other indices that assess vulnerability at different levels have been developed as follows: The Global Notre-Dame (ND-Gain) adaptation index [11] which also has components of exposure, sensitivity, and adaptive capacity from which this current index was developed. However, it differs from the current in that it is broader and covers other sectors of vulnerability such as health, ecosystems, habitat, water, and infrastructure. Furthermore, the crop-drought indicator which provides indicators of vulnerability to droughts [12], the water-poverty index which shows vulnerability in the context of accessibility to water based on poverty [13] and the farmer vulnerability to global change index which focuses on large scale irrigation and maladaptation and associated vulnerability [14] (Table 1).

It is important to note that of all these indices, the ND Gain index is one of the best-known vulnerability indices. It simulates a country's current vulnerability to climate change disruptions and assesses the country's readiness to leverage private and public sector investments for adaptive purposes. This index is one of the gold standards of creating vulnerability indexes. It has a component that assesses readiness for adaptation based on social, economic and governance. This aspect is far from our own index which essentially focuses on crop vulnerability and its three measures within an agricultural context. In creating this current index, the initial study focused on the food sector aspect of the ND Gain index and does not simulate the other sectors identified by ND Gain. Just like in our own vulnerability index, the ND Gain uses proxies for adaptive capacity, and estimates exposure, and sensitivity (Table 1).

Approaching vulnerability as a function of sensitivity, exposure, and adaptive capacity can be rationalized as follows, 1) It incorporates an integrated approach which further enhances a holistic perspective. In the context of this study, the vulnerability index reflects the role of historical crop yields, historical precipitation, and socio-economic proxies of adaptive capacity [32, 33]. 2) Secondly, the approach adopted also enhances the quantification of variables such as the adaptive capacity that is difficult to represent. In most of Africa, the difficulty of having data on adaptive capacity has made estimating it daunting and has made scientists to resort to the use of proxies. This study uses the two most important proxies of adaptive capacity in Cameroon which are literacy and poverty rates. 3.) Thirdly, this approach conceptualizes vulnerability in a more fundamental and practically useful fashion. The results of the assessment and the indicators selected to assess vulnerability can be analysed to identify the drivers of vulnerability. Addressing the drivers of vulnerability provides a reliable approach to reduce the current vulnerability and manage potential risks [33].

## 2.2. Sensitivity and exposure indices

The sensitivity index heralds the measurable decline in crop yields due to climate stressors and extreme events [2, 27–29]. In other words, sensitivity is the manifestation of climate change, climate variability, and extreme events on crop yields. Therefore, the sensitivity index focuses on the actual yield responses to growing season precipitation. Theoretically, the sensitivity index (crop vulnerability) has a direct relationship with the exposure index (drought vulnerability) mainly through evapotranspiration. This is seen as crops might carry out transpiration at varying rates thereby affecting the amount of moisture lost to the climate system or may shade the ground depending on their canopies and thereby affecting evaporation [2]. Also, the roots of crops might enhance evaporation by making the soil porous and exposing it to evaporation [2, 28, 29]. It is important to emphasise here that this current study does not focus on the aspects of causality between these variables but simply describe the theoretical relationships that is observed. The exposure index further describes the extent and nature of the stimulus reflected in the magnitude, intensity, and duration of the moisture deficit [2, 27–29] (Fig 1). The exposure index focuses on the actual precipitation responses to changes in climate. These two sub-indices tend to have a direct relationship with vulnerability index which implies causality and an inverse relationship with the adaptive capacity index which does not imply causality. This relationship is inverse because when these two sub-indices are high the adaptive capacity index is low because the concerned farmers are unable to adequately respond/cope with the yield and precipitation declines due to low adaptive capacity. In other words, farmers who are less likely to adapt due to low adaptive capacity are more likely to suffer from the effects of declining yields and precipitation. The observed inverse relationship does not imply causality. Low adaptive capacity index can only indirectly impact exposure and sensitivity indices by affecting the ability of the farmers to adapt to changes in exposure and sensitivity. This can be reflected in land use practices which maladapt the farmers to the changes in exposure and sensitivity.

## 2.3. Adaptive capacity index

The adaptive capacity index depicts the ability of a production system to adjust or adapt or cope with multiple stressors and shocks including climate change, climate variability, and extreme events [2, 28, 29]. When linked to this study, the intensity of the effects of droughts on crop yields is often mitigated by the adaptive capacity of the farmers producing the crops to manage the drought. This adaptive capacity is often reflected in proxies such as literacy and poverty rates (Fig 1). The socio-economic proxies of adaptive capacity are the level of literacy and poverty which are safety nets in the face of climate stressors [34, 35]. The adaptive capacity index does not affect the exposure, sensitivity, and vulnerability indices directly; it does so indirectly through its proxies such as literacy and poverty rates. However, the adaptive capacity index has an inverse relationship with the vulnerability index, exposure index, and the sensitivity index. This inverse relationship does not imply direct causality as adaptive capacity can only impact these other indices indirectly. When the adaptive capacity index is low, it means the farmers concerned have lower literacy levels and higher poverty levels and this is often associated to a higher exposure index reflected in a limited ability to adapt. Also, a low adaptive capacity index is associated to a higher sensitivity index which is reflected in the limited ability of the farmers to invest in options that will enhance crop yields such as high yielding varieties, fertilizers or even tractors. Furthermore, a low adaptive capacity index is associated with a high vulnerability index; this is also judged by the prevalence of higher poverty rates and low literacy rates which are associated with higher vulnerability due to increase exposure and limited ability to cope with precipitation and yield deficits. All these sub-indices (sensitivity,

exposure, and adaptive capacity) together lead us to what is the vulnerability index. It is important to observe that the adaptive capacity index is an exogenous index that is influenced by external explanatory variables such as literacy and poverty rates.

## 3. Materials and methods

### 3.1. Study site

Cameroon is a country in Central Africa; in 2019 its population was estimated at about 25.88 million people [36]. The country is located between latitudes 1.7° N-13.8° N and longitudes 8.4° E-16.8° E [37]. Cameroon is bordered in the south by Equatorial Guinea, Gabon, and Congo, to the east by Central African Republic and Chad, to the north by Chad and to the west by Nigeria. Agriculture is the "life wire" of the Cameroonian economy as it employs about 65–70% of the population, contributes about 52% to the country's GDP and 45% to Cameroon's export earnings and 15% to public revenue [37]. An important characteristic of agriculture in Cameroon is that it is rainfed and essentially in the hands of small-scale peasant farmers.

The southern part of Cameroon has the humid forest agroecological zone. This is further balkanized as follows: the southern-central-eastern part of the country with a humid forest and a bi-modal precipitation of about 1600 mm annually, in the south-west with its humid forest and unimodal/mono-modal precipitation of about 3000 mm annually, the western highland in the north-western part of the southern portion of the country with an annual precipitation of about 2000 mm. In the northern parts of the country is found the High Guinea Savana around the Adamawa Region with an annual precipitation of about 1200 mm and the Sudano-Sahelian region that stretches up to the Lake Chad region with annual precipitations dropping to an estimated 500 mm. While it can be observed that precipitation declines from the south to the north of the country, there are also changes in crops that grow in different parts of the country [37]. In the south, we generally have tubers and grains while in the north, the agroecological properties favor the cultivation of mostly grains/cereals (Fig 2). However, in Cameroon maize is cultivated in both the south and north of the country while millet is cultivated essentially in the north (Adamawa, North and Far North Regions). The most important zones of rice cultivation are the western highlands (North West and Western regions). However, rice is also cultivated in smaller quantities in other regions in the north of the country (North and Far North Regions) as well as in smaller areas in the centrerRegion, the south east and east regions. In general, most of these crops are cultivated in the often dry and drought stricken northern parts of the country. Unfortunately, due to data availability at the sub-national scale, this work has only been validated at four sub-national sites/scales. The sub-national sites selected are consistent with the areas in which these crops are mostly cultivated and where data are available.

### 3.2. Data collection

The principal spatial dimension of this study is essentially national and validated at four sub-national scales based on data availability. To accomplish the objectives of this study, data on the various sub-components of vulnerability were collected. To compute the sensitivity indices to droughts at the national scale, time series data on observed crop yield for all three crops (maize, millet, and rice) for the period 1961–2019 in hg/ha/year were collected from FAO-STAT [38]. To validate the national scale sensitivity index, sub-national scale crop yield data (hg/ha/year) were found for four sites in Cameroon, these include: southern maize, northern maize, western highland rice, and northern millet. These sites are consistent with the zones in which these crops are predominantly cultivated. Unfortunately, the time series is only available for the period 1991–2016 [38].

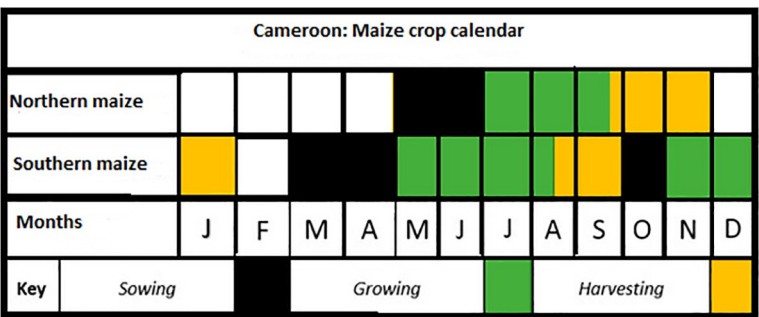

**Fig 3. Maize crop calendar for Cameroon.** Source: Authors' conceptualization.

The exposure indices of these crops to droughts for the national scale analysis were computed from the mean annual precipitation data and specific mean crop growing season precipitation data for the period 1961–2019 for each crop. These station-based data were collected from the collaborative database of the University of Oxford and UNDP [39–42] and the Climate Change Knowledge Portal of the World Bank Group [36] (https://climateknowledgeportal.worldbank.org/). Since the growing seasons of these crops are different; crop calendars for these crops were inspired from [43, 44] The final mean crop growing season calendars for rice, maize and millet in Cameroon are presented in Figs 3–5. Therefore, the mean crop growing season precipitation data for each year was a mean aggregate for the growing season months of that year. To validate the national scale exposure index, sub-national mean annual precipitation and mean annual crop growing season precipitation data were downloaded from the Climate Change Knowledge Portal of the World Bank Group [36] (https://climateknowledgeportal.worldbank.org/). The four sites in which maize, millet and rice are mostly cultivated in Cameroon include: southern maize, northern maize, western highland rice, and northern millet. The time series data were collected for the period 1991–2016, consistent with the period in which crop yield data are available. It is important to note that maize has a bi-modal growing season in the south and a uni-modal growing season in the north of the country. As regards to millet, there is a uni-modal growing season in the north and central part of the country. In the case of rice, both the south and the western highlands have a uni-modal growing season (Figs 3–5). 'Theoretically, the areas with bi-modal growing season are likely to have lower vulnerability, sensitivity, and exposure indices while the reverse is true for the uni-modal areas.

The adaptive capacity index used in this study is based on two proxies which include: poverty (%) (material asset) and literacy rates (%) (human asset). The poverty rate includes both

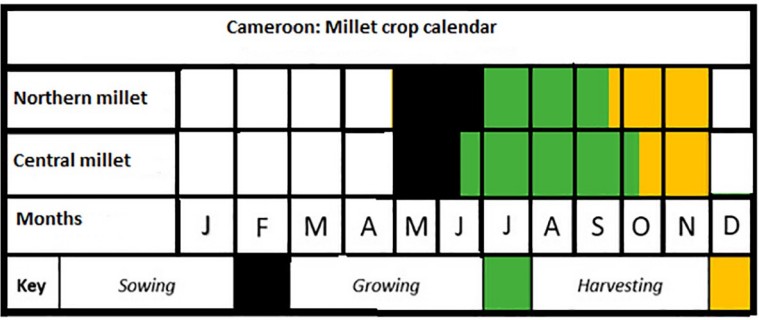

**Fig 4. Millet crop calendar for Cameroon.** Source: Authors' conceptualization.

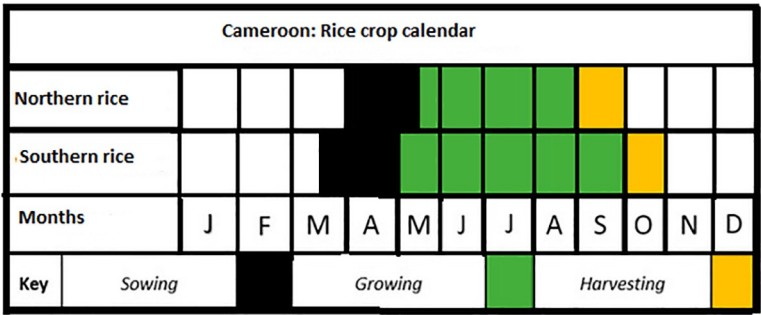

**Fig 5. Rice crop calendar for Cameroon.** Source: Authors' conceptualization.

material and financial assets which are complete reflections of a people's ability to adjust to climate shocks by building resilience and adopting alternative livelihood sustenance options. The poverty rate data and literacy rate data for the national scale simulations were collected for the period 1961–2019 while those for the four sub-national scale sites were collected for the period 1991–2016 from the African Development Bank [45] (https://www.afdb.org/en/countries-central-africa-cameroon/cameroon-economic-outlook) and [46, 47] (https://knoema.com/atlas/Cameroon/topics/Education/Literacy/Adult-literacy-rate) respectively. These two socio-economic proxies are among the best that describe the variations in adaptive capacity and how these might impact resilience to climate shocks in predominantly crop production economies like Cameroon.

Finally, the vulnerability index was based on the data collected for all the three indices described above. This includes: the sensitivity, exposure, and adaptive capacity indices. This approach is mandated by the fact that vulnerability is usually a function of the above three indices. For details on how this was simulated, see the section on data analysis below.

### 3.3. Data analyses

**3.3.1. Vulnerability index.** To compute the empirical vulnerability index at the national scale and sub-national scale, this study uses a composite vulnerability index which was first developed by [5]. This index is like other vulnerability indices such as the ND-GAIN [11], the crop-drought indicator [12], and the water-poverty index [48–50]. This current index is unique because it is developed specifically for application in an African crop farming context. This index captures crop-specific parameters such as yield, precipitation, and the adaptive capacity of the farmers. All these computations were performed in excel and SPSS version 20. The equation used to compute this composite index is given below (Eq 1).

$$VU_{xinsn} = SE_{xinsn} + EX_{xinsn} - AdC_{xinsn} \qquad (1)$$

where $VU_{xinsn}$ is the crop yield vulnerability index for the selected crops (rice, maize, and millet) at both national and sub-national scale, $SE_{xinsn}$ is the crop yield sensitivity index at both national and sub-national scale, $EX_{xinsn}$ is the crop yield exposure index at both national and sub-national scale and $AdC_{xinsn}$ is the crop yield adaptive capacity index at both national and sub-national scale. Also, *x is* the year and *i* is a given crop.

**3.3.2. Sensitivity index.** To compute the sensitivity index, the time series historical observed crop yield data for the national scale analysis for the period 1961–2019 and the sub-national scale analysis for the period 1991–2016 were subjected to detrending. This involves removing a linear model in the time series data by dividing the projected linear trend by the observed linear trend using Eq 2 below. The advantages of detrending are its ability to remove

the repercussions of increased technology, ability to illustrate annual crop yield variations driven by precipitation and reduce the effects of consistency errors in reporting [5, 17, 30, 51]. The trend line equations for a simple linear regression (Eqs 3, 4 and 5) were used to project and obtain the projected or expected crop yields for each year for each of these crops. The observed and projected crop yield data were regressed against the precipitation data. The trend line equation is based on the ordinary least squared method. The sensitivity indices for the various crops were fitted by dividing the mean expected crop yields by the mean observed crop yields (Eq 6). This approach is likened to that used by previous studies [5, 12, 55]. The polynomial model was subsequently used because the trends of rice were non-linear and therefore to better optimize the model a polynomial model was selected.

$$EXP_{ynsn} = ax + b \tag{2}$$

where $EXP_{ynsn}$ is the expected crop yield at both the national and sub-national scale, $\chi$ is the year, $a$ is the linear trend, $b$ is the intercept when $EXP_{ynsn} = ax$.

$$EXP_{ymaizen} = 258,56x + 7144,4, \ R^2 = 0.63 \tag{3}$$

$$EXP_{ymilletn} = 122,8x + 6146,8, \ R^2 = 0.75 \tag{4}$$

$$EXP_{yricen} = 48,95x + 19902, \ R^2 = 0.0039 \tag{5}$$

where $EXP_{ymaizen}$, $EXP_{ymilletn}$, $EXP_{yricen}$ represents expected or projected maize, millet, and rice yields in Cameroon at the national scale; $x$ represents the years while the numbers in the equations are the slopes and intercepts from left to right respectively.

$$SE_{xy}nsn = \frac{EXP_{yxynsn}}{ACT_{xy}nsn} \tag{6}$$

where $SE_{xynsn}$ is the crop yield sensitivity index at both national and sub-national scale, $EXP_{xynsn}$ is the mean projected or expected crop yield at both national and sub-national scale, $ACTx_{ynsn}$ is the mean actual crop yield at both national and sub-national scale. $y$ represents the yield and $x$ the years notation.

**3.3.3. Exposure index.** The exposure index was now computed by dividing the mean annual monthly precipitation for the period 1961–2019 for the national scale analysis and for the period 1991–2016 for the sub-national scale analysis by the mean growing season precipitation for each crop during the period 1961–2019 for the national scale analysis and 1991–2016 for the sub-national scale analysis. This is like methods used in other studies [5, 12, 52]. The growing season precipitation better reflects the actual growing conditions for the crop [59]. Eq 7 below was used to fit the exposure index.

$$EX_{xpnsn} = \frac{\mu LT_{annualpptnsn(1961 \ to \ 2019)/1991-2016}}{\mu STY_{growingSpptnsn(1961 \ to \ 2019)/1991-2016}} \tag{7}$$

where $EX_{xpnsn}$ is the crop yield exposure index at both national and sub-national scale, $\mu LT_{annualpptnsn(1961 \ to \ 2019)/1991-2016}$ is the mean annual precipitation at both national and sub-national scale, $\mu STY_{growingSpptnsn(1961 \ to \ 2019)/1991-2016}$ is the mean growing season precipitation at both national and sub-national scale. $x$ and $p$ represent years and precipitation respectively.

**3.3.4. Adaptive capacity index.** Adaptive capacity index (see Eq 8) was parameterized based on two proxies: poverty and literacy rates. These two proxies were selected because data on the other potential proxies, such as route network, safety nets, natural resources

are limited. Also, these two proxies adequately capture adaptive capacity and affect all other proxies. For example, poverty reduction can lead to improvements in the literacy rates (human assets) and the spill over effects of this could be reflected in improved social connections, networks, and safety nets (social assets). Though these proxies have an inverse relationship, they are not entirely independent. Reducing poverty can enhance literacy rates and improve resilience to climatic shocks through climate knowledge accessibility and other means to invest in more resilient cropping systems. The equation used is presented below (Eq 8).

$$AdC_x nsn = \left( \frac{10^2 - P_{xt} nsn}{10^2} \right) + \left( \frac{L_{xtnsn}}{10^2} \right) \tag{8}$$

where $AdC_{xnsn}$ is the crop yield adaptive capacity index at both national and sub – national scale, $P_{xt}nsn$ is the poverty rate (%) at both national and sub-national scale, $L_{xt}nsn$ is the literacy rate (%) at both national and sub-national scale and $x$ represents the year notation.

## 4. Results

### 4.1. Recent national scale trends in maize, millet and rice yields in Cameroon

In assessing the observed yields of maize, millet, and rice for the period 1961–2019, the following observations can be made. Firstly, this study shows that maize and millet have consistently increased over time with the exceptions of some blips. For example, between 1961–2019, maize yields increase by 150% from above 6000 (hectograms per hectare: hg/ha) in 1961 to slightly above 15000 hg/ha in 2019 (Fig 6A). This increase was occasioned by blips with the most prominent being a recurrent decline from the year 2000 from about 25000 hg/ha to 16264 hg/ha in 2019, a 34% decline. Millet on the other hand also witnessed an increase of 55.56% from above 9000 hg/ha in 1961 to about 14000 hg/ha in 2019 (Fig 6B). Even though millet witnessed declining blips in 1986, 1990 and 2008, the subsequent yields showed recovery and continuous increase in production. Rice on its part witnessed an increase from about 4800 hg/ha in 1961 to about 56000 hg/ha in 1989. Subsequently, the crop tilted towards a consistent decline ending at about 11664 hg/ha in 2019 (Fig 6C).

For the three crops described above, millet has the highest coefficient of determination ($R^2$) of about 79%. Maize has an $R^2$ of about 71% while rice has and $R^2$ of about 54%. With an $R^2$ of about 79%, this study has found that in the context of millet, the model and data used explains about 79% of the variability of the response data around its mean (Fig 6B). This further supports the argument that millet is more resilient, less vulnerable and is relatively experiencing rising yields when compared to maize and rice. Maize on the other hand also has an $R^2$ of about 71% which also indicates that the model explains about 71% of the variability of the response data around the mean (Fig 6A). This supports the argument that maize is less resilient and more vulnerable to droughts when compared to millet but also more resilient and less vulnerable than rice. With an $R^2$ of 71%, the relatively increasing trend in maize yields can be better understood. Rice has the lowest $R^2$ of 54% which reflects lower resilience and high vulnerability when compared to maize and millet (Fig 6C). Also, the very low $R^2$ indicates and confirms the relatively declining rice yields observed in the last half of the historical period covered by the data. In addition to millet having the highest $R^2$, it also has a higher slope and intercept when compared to the other crops. Rice on it part has the lowest $R^2$ as well as slope and intercept.

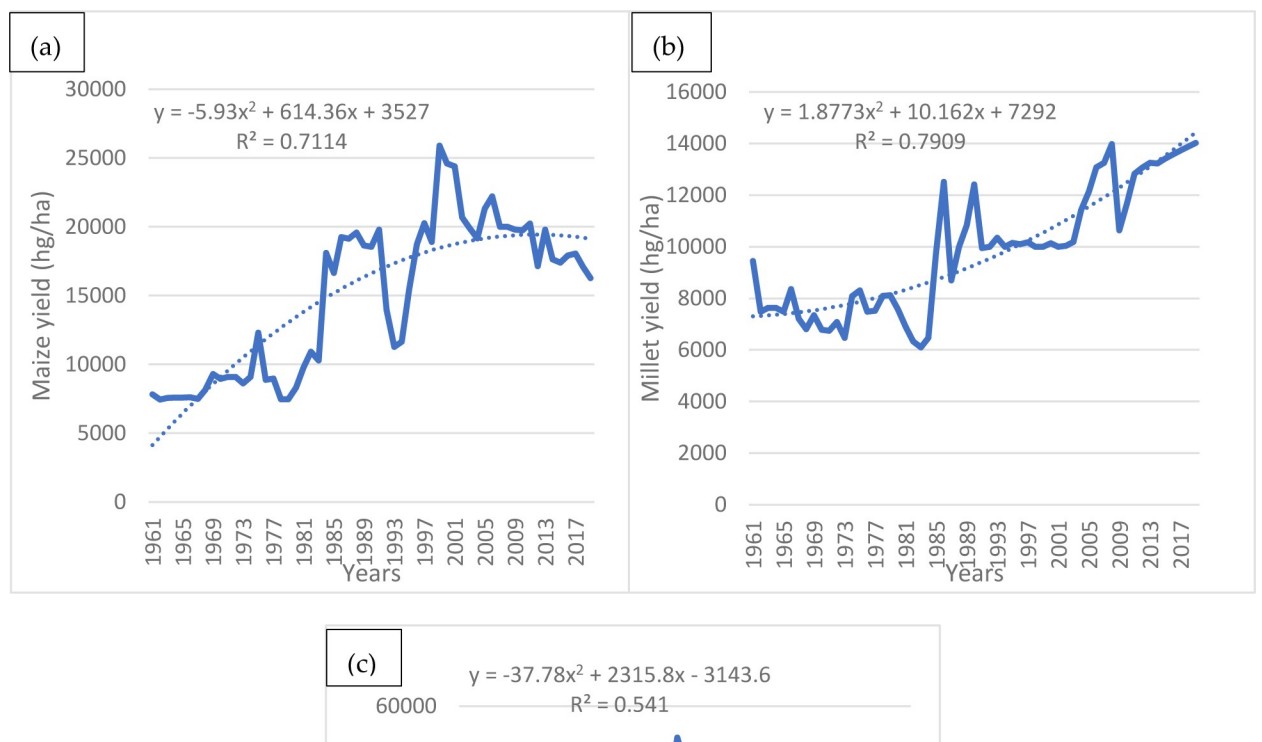

**Fig 6.** National scale observed (a) Maize yield, (b) Millet yield and (c) Rice yield for Cameroon based on a polynomial model. Source: Authors'
conceptualization.

## 4.2. Recent national scale trends in maize, millet and rice growing season precipitation in Cameroon

Results for the observed trends in growing season precipitation for the three crops show some variations. The mean growing season precipitation for millet straddles around 200 mm (Fig 7B) while that for maize straddles around 175 mm (Fig 7A). Rice which experiences a declining trend is less resilient, more vulnerable, and has a mean precipitation of about 200 mm (Fig 7C), much higher than that for maize. It is important to note that this depicts the fact that the different crops have different growing seasons and as a result the mean growing season precipitation for rice might be higher than that for maize, yet rice is still vulnerable. Secondly, this

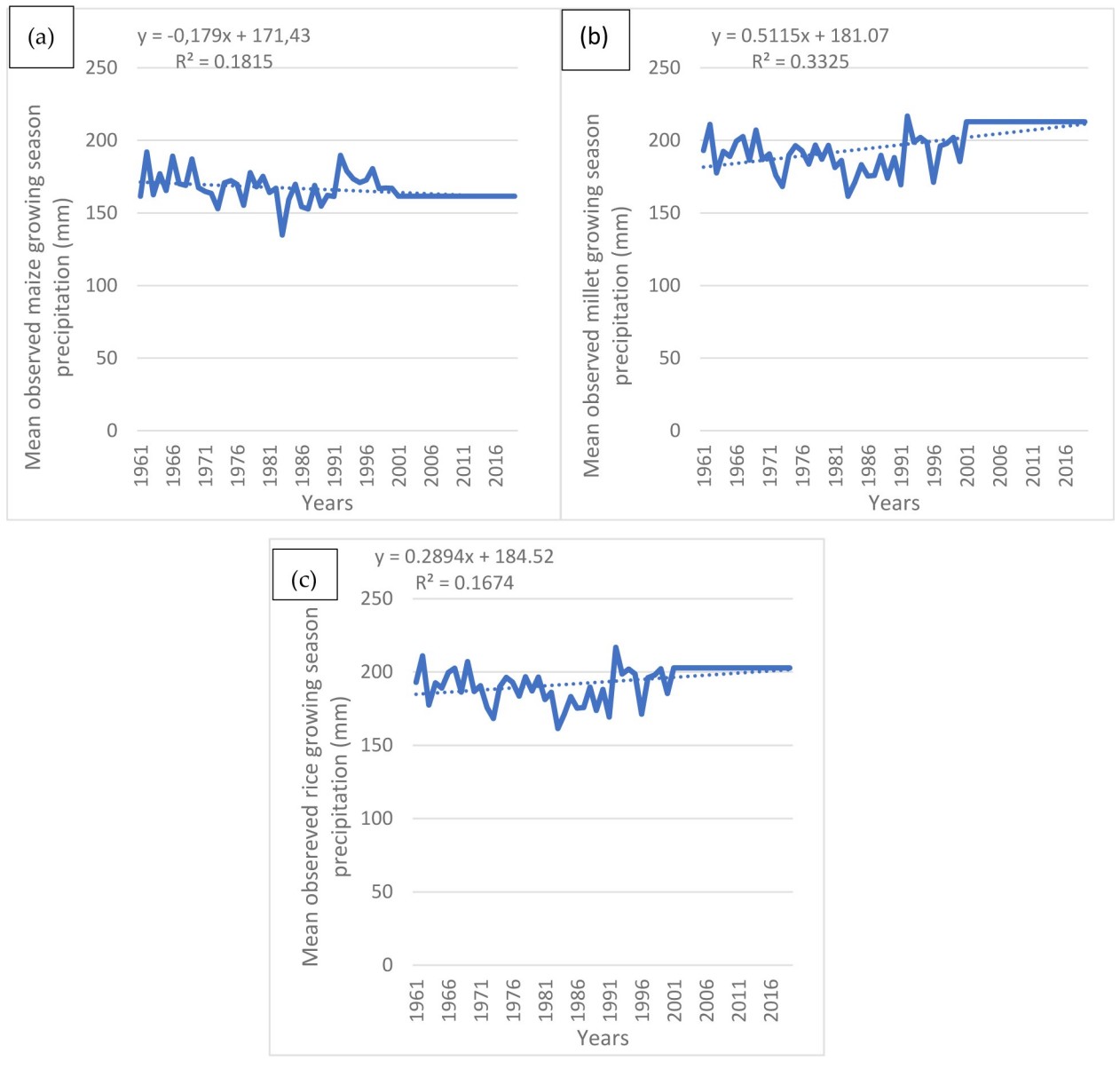

**Fig 7.** National scale observed (a) Maize growing season precipitation, (b) Millet growing season precipitation and (c) Rice growing season precipitation for Cameroon. Source: Authors' conceptualization.

underscores the fact that even though precipitation levels are significant moisture requirement determinants for any crop, other variables are often needed to create a better understanding. In addition, this is also dependent on the distribution of the precipitation and the occurrence of outliers.

It is important to note that the growing season precipitation averages about 200 mm because it is based on: 1. Only the mean for the growing season months of each crop which varies and is often shorter than the annual values, 2. Also, the growing season precipitation used here reflects mostly data based on an aggregate mean that reflects all the crop calendars across the country and 3. Most of these grains are cultivated in the northern regions of Cameroon which are often hit by recurrent droughts and water scarcity (see areas of cultivation in

the section on study area). From 2001 to 2019 the mean growing season precipitation was flat for each of the years thereafter for maize, millet, and rice. To the best of our knowledge, the growing season precipitation assumed a flat state for these crops due to a reduction in the frequency of perturbations such as droughts and the occurrence of more stable growing season precipitation. However, though flat, this does not mean reliable as this often falls below the minimum threshold required to sustain crops and avoid droughts.

### 4.3. National scale maize, millet and rice yield vulnerability and adaptive capacity indices

From the computed vulnerability indices, this study has found that of the three crops, rice has the highest vulnerability index of 0.77 (Fig 8A) while maize has the second highest vulnerability index of 0.64 (Fig 8B, Table 2) and millet has the lowest of 0.5. This implies millet is the

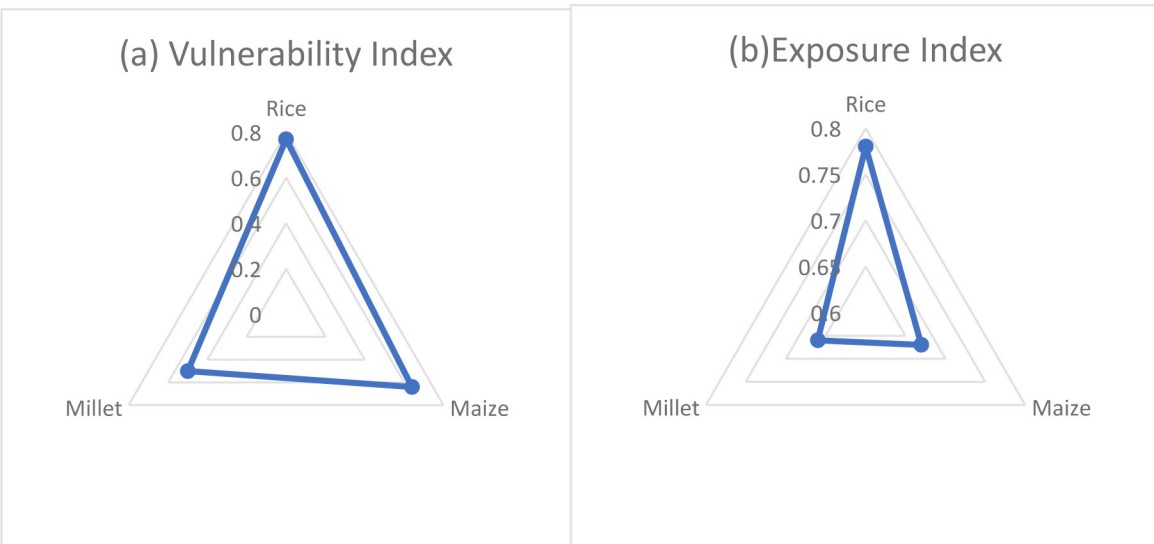

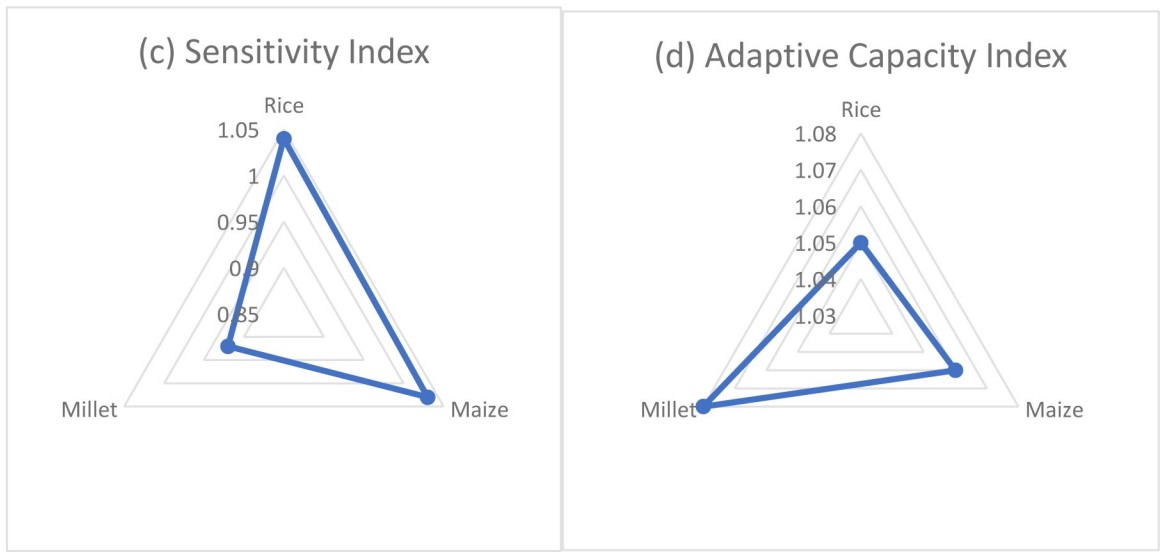

**Fig 8.** Simulated national scale (a) Vulnerability Index, (b) Exposure index, (c) Sensitivity Index and (d) Adaptive Capacity Index for Cameroon. Source: Authors' conceptualization.

**Table 2. National scale vulnerability indices and R² of the linear and polynomial models.**

| Crops | Vulnerability Index | Exposure Index | Sensitivity Index | Adaptive Capacity Index | R² of linear model | R² of polynomial model |
|-------|---------------------|----------------|-------------------|-------------------------|--------------------|------------------------|
| Rice | 0.77 | 0.78 | 1.04 | 1.05 | 0.0039(0.39%) | 0.54(54%) |
| Maize | 0.64 | 0.67 | 1.03 | 1.06 | 0.63(63%) | 0.71(71%) |
| Millet | 0.5 | 0.66 | 0.92 | 1.08 | 0.75(75%) | 0.79(79%) |

Source: Authors' conceptualization.

least vulnerable while rice is the most vulnerable of the three crops under investigation (Fig 8C, Table 2). This is consistent with the initial trends in the yields of these crops described above which shows that millet and maize are witnessing increasing yields while rice is declining. Also, millet has the highest R² followed by maize. The exposure and sensitivity indices are also consistent with the vulnerability index. In other words, when the vulnerability index is high, the exposure and sensitivity indices are also high. As can be seen on Figs 8B–8D and 9, millet has the lowest exposure and sensitivity indices of 0.66 and 0.92 respectively while the exposure and sensitivity indices for maize and rice are much higher. Rice on the other hand also has the highest exposure and sensitivity indices among all three crops (Figs 8B, 8C and 9 and Table 2).

With regards to the adaptive capacity index, this study has found that millet has the highest adaptive capacity index (1.08) followed by maize (1.06) while rice (1.05) has the lowest adaptive capacity index (Fig 8D, Table 2). Though the margin of difference between these indices is not very high, rice still stands out as the least adaptive of the three crops. The interesting observation here is that the crops with the highest adaptive capacity indices also have the lowest vulnerability, sensitivity, and exposure indices. An inverse indirect relationship exists between the adaptive capacity index and vulnerability, exposure, and sensitivity indices.

## 4.4. Sub-national vulnerability index model validation

Due to issues of data availability, the sub-national analyses were performed only for sites in which data were available; these sites coincide with the sites in which these crops are most

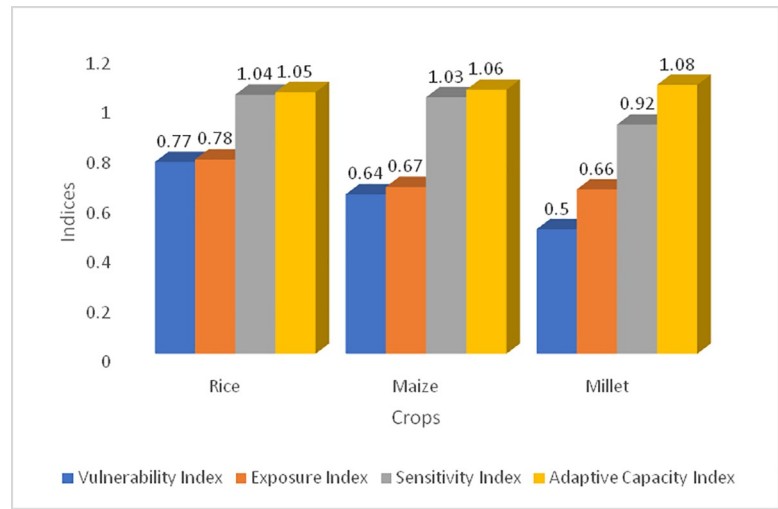

**Fig 9. Simulated national scale vulnerability, exposure, sensitivity, and adaptive capacity indices.** Source: Authors' conceptualization.

**Table 3. Sub-national vulnerability indices at four sub-national sites.**

| Sub-National Crops | Vulnerability Index | Exposure Index | Sensitivity Index | Adaptive Capacity Index |
|---|---|---|---|---|
| Southern Maize | 0.51 | 0.55 | 1.02 | 1.06 |
| Northern Maize | 1.21 | 1.13 | 1.04 | 0.94 |
| Western Highland Rice | 0.78 | 0.76 | 1.04 | 1.02 |
| Northern Millet | 0.52 | 0.68 | 0.89 | 1.05 |

Source: Authors' conceptualization.

dominantly cultivated. Since maize grows in both the north and south of the country, this study investigated both southern and northern maize to establish a spatial variation. The results show that northern maize has the highest vulnerability index of about 1.21 while southern maize has a vulnerability index of 0.51 (Table 3). The higher vulnerability index of northern maize is linked to the fact that the north of the country witnesses recurrent and frequent droughts caused by declining precipitation from south to north. On the other hand, southern maize has the highest adaptive capacity index of 1.06 while northern maize has the lowest adaptive capacity index of 0.94 that is consistent with the high vulnerability index (Table 3). In the context of maize, it can be said that the vulnerability increases from the south to the north. The second most vulnerable crop of the three is western highland rice with a vulnerability index of 0.78 and the second smallest adaptive capacity after northern maize of 1.02 (Table 3). The relatively high vulnerability index and low adaptive capacity index of western highland rice at the sub-national scale is consistent with the high vulnerability index recorded for rice at the national scale. Northern millet is the second least vulnerable crop after western highland rice with a vulnerability index of 0.52 (Table 3). These results shows that if northern maize is left out, western highland rice is the most vulnerable crop.

In terms of growing season precipitation (Fig 10), it can be observed that southern maize has the highest growing season precipitation followed by western highland rice. Northern millet and northern maize have the same growing season, the reason why they both assume the same rates. It can be drawn from this figure that the growing season precipitation also declines from the south of the country to the north as northern millet and maize have the lowest growing season precipitation records. The records show that the mean growing season precipitation for southern maize is about 300 mm annually while that for the western highland rice is about 200 mm annually. Northwards, northern maize, and millet record a mean growing season precipitation of about 150 mm annually. This confirms the hypothesis of declining precipitation northwards and therefore increased constraints on yields.

## 5. Discussion

These results show that at the national scale rice has the highest vulnerability index and the lowest adaptive capacity index. However, the growing season precipitation for rice in Cameroon is relatively higher than the growing season precipitation for maize. Normally, it is expected that with a higher mean growing season precipitation than maize, rice should not be the most vulnerable crop; but this is not the case. This still however makes sense as it is important to consider that these crops all have different crop production seasons (see Figs 3–5) and are impacted by several other factors other than growing season precipitation. In addition, it is not only the mean annual growing season precipitation that is important in crop growth but rather growing season precipitation that is well distributed throughout the growing season of the crop. If the precipitation is concentrated at the beginning, middle or end of the season, then it is likely that crop growth might be affected negatively. In most cases, adequate growing

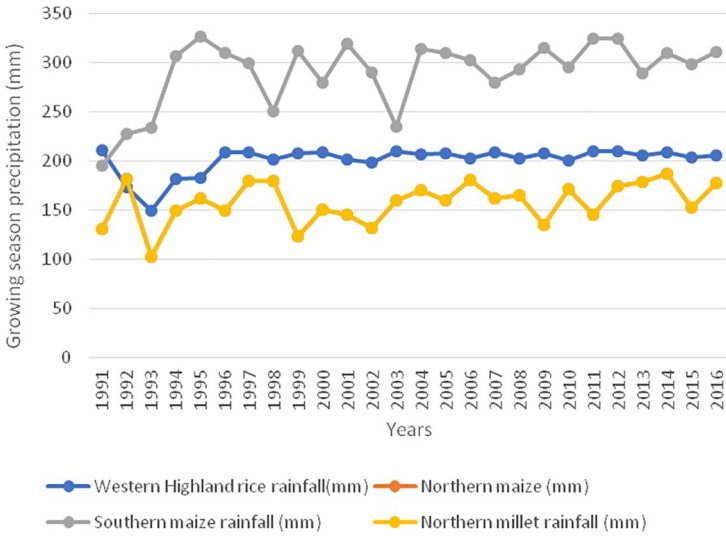

**Fig 10. Sub-national growing season precipitation trends for Western Highland rice, Southern maize, Northern maize, and millet.** \*\*northern maize and millet are computed from the same growing season precipitation, reason why they are identical i.e., growing season for Northern millet equals growing season for northern maize. Source: Authors' conceptualization.

season precipitation must be associated with moderate temperstures to ensure good harvest. Even at this rate, when there is adequate precipitation and temperature, crop failures might still occur if the soil is of poor quality, the planting species are not resistant to droughts or crop pests and the farmers do not have adequate access to organic and inorganic inputs. Therefore, as can be seen here crop vulnerability is a function of several variables and not just entirely climatic.

The observations of declining rice yields in Cameroon that began around 1989 are consistent with the observations of previous studies. It has been argued that rice production in Cameroon was stable between 1960–1985. This period equally experienced spectacular evolution marked by increases in rice in terms of the relationship between yields and cultivated area. However, due to declining rice prices in the 1990s triggered by a decline in profitability, the economic crisis, and the devaluation of the local currency in 1994 the country tilted from being self-sufficient in rice production to being a net importer of rice [53–55]. In fact, the production trends identified by [39] indicate an increase up to 1985 and a decline thereafter. This period of net decline in production was also consistent with the decline in support of irrigated rice production, a decline in fertilizers, pesticides and use of tractors. Furthermore, the fact that rice cultivation is in the hands of small-scale farmers (93%) who are ill equipped to meet the production challenges also comes to play. Consequently, in 2005, production was only able to supply 13% of Cameroonian rice consumption [53, 54]. This is similar to what obtains in most sub-Saharan African countries where small scale agriculture remains the dominant economic activity [56]. Also, investments in small scale agriculture will not necessarily yield higher yields as the changes that will affect small scale farming such as technology, markets, climate, and global environment will be affected by the huge heterogeneity and different national priorities [56, 57]. To further support this view, Shiferaw et al. [58] observe that the agrarian economies of sub-Saharan Africa are extremely sensitive to climate variability. The major form of this variability is through droughts that have been a major source of malnutrition and famine. Like in Cameroon, the impact of a drought depends on economic, social, and

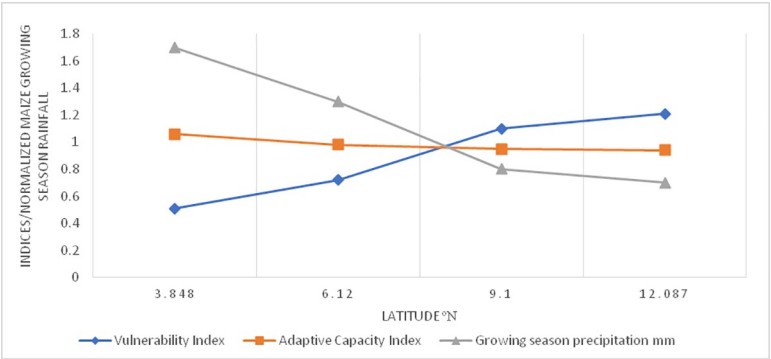

**Fig 11. Latitudinal evolution of sub-national maize vulnerability indices, adaptive capacity indices and normalized maize growing season precipitation in Cameroon.** Source: Authors' conceptualization.

environmental impacts of the droughts. Major potential options for adaptation recommended are policy and agricultural technology [58].

Furthermore, this study has shown that there is an inverse indirect relationship between vulnerability and adaptive capacity. This is also consistent with other studies across Africa. For example, a study conducted on the vulnerability of maize yields to droughts in Uganda in 2017 reported that there is a latitudinal variation in vulnerability with high levels of vulnerability of maize being linked to the higher latitudes in the north of the country while lower rates are observed in the south. At the same time, the study shows that the northern regions of Uganda have lower adaptive capacities when compared to the south of the country [5]. However, this current study has also observed the same trends with respect to southern and northern maize. It can be seen that southern maize has a lower vulnerability index and a higher adaptive capacity index but in the north the vulnerability index of northern maize is higher while the adaptive capacity index is lower (Fig 11). In addition to variations in precipitation distribution during the maize growing season, other drivers of these variations are soil quality, and socio-economic status of the farmers involved in cultivating the crop [5]. Furthermore, another study that focused on the vulnerability of maize, cassava, beans, millet, groundnuts, and sweet potatoes agrees with the inverse indirect relationship between vulnerability and adaptive capacity. This study found cassava to be the most vulnerable crop at the national scale and maize the least vulnerable in Uganda [10]. Also, the fact that different crops are vulnerable across West-Central (Cameroon) to East Africa (Uganda), indicates that a movement from one ecological zone to another is associated with changes in levels of vulnerability as well as in the types of crops that are vulnerable due to differences in climate, soils, and other socio-economic conditions (Fig 12).

Actions must be taken to enhance and revamp rice production in Cameroon. Much can be done through the introduction of high yielding varieties, drought and pests tolerant and resistant crop types, soils improvements and climate information management. Varieties of rice that do not have a well developed aerenchymal are capable of sustaining rice yields while reducing emission of methane [59, 60]. This is very important for rice production because the country depends a lot on rice imports and it is one of the most consumed crops in SSA and Cameroon [5, 61]. It has been argued that since most of the rice cultivated in Cameroon is paddy rice, investments in modern irrigation infrastructure will go a long way to enhance production. In fact, Goufo [61], argues that one of the main problems with rice production in Cameroon has to do with deteriorating irrigation infrastructure. To mitigate this effect, synergies between the government, farmers, NGOs, and other stakeholders must be put in place to

install new and efficient irrigation facilities such as dams, canals, and drainage channels. However, this must be accompanied by improvements in access to agricultural production inputs such as fertilizers, pesticides, and tractors. All these efforts will be useless if the practice in which the Bororo herdsmen use the rice valleys for four months a year to feed their herds between March and November which is part of the crop growing season [61] is not controlled. Even though irrigation is said to enhance production, water management through mid-season drainage increases oxidation of soils and reduces methane emissions. However, water should only be reduced to a level that can be supported by the crops and this can be done daily [47, 61] (Fig 12). Water levels should later be re-established to avoid prolonged water stress and consequent poor yields [61].

While the animals might be dangerous for the crops because they might eat the crops and damage them, this study is suggesting a mixed farming model in which the animals can be kept in different parts of the farms and fed with the rice straws from the paddy fields and the animal dung is used as fertilizers for the crops [59, 60]. Organic materials like animal dung have the beneficial potential of enhancing soil aeration and reducing methane emissions. Other additives like sulfates can also reduce the amount of methane emitted from paddy rice fields. Caution must be taken to ensure intermittent draining of the fields to reduce anerobic bacteria and thus methane missions which often increase with increasing flooding [61] (Fig 12).

At the sub-national scale, northern maize has the highest vulnerability index and the lowest adaptive capacity index. This is consistent with the recurrent droughts and persistent declines in precipitation from the south to the north (Fig 11). On the other hand, the fact that southern maize has the lowest vulnerability index is explained by the wetter nature of the area that contrasts with the north of the country. These finding show that different crops are vulnerable at the national and subnational scale. This is important to underscore because what is obtained at the national scale might not be reflected at the sub-national scale as different regions are differentially affected by different precipitation characteristics, temperatures, soils, access to farm inputs, high yielding varieties and crop pests [6, 47, 61]. Therefore, just like with national scale rice, northern maize will benefit from a combination of irrigation and farm inputs in the context of organic fertilizers and enhanced farmer access to climate related crop planting information to better guide farming decisions.

## 6. Conclusion

This study has found that at the national scale and for the three crops under study, millet has the lowest vulnerability index while rice has the highest vulnerability index. Sub-nationally, northern maize is the most vulnerable crop followed by western highland rice. There is an indirect inverse relationship between vulnerability and adaptive capacity at both scales of

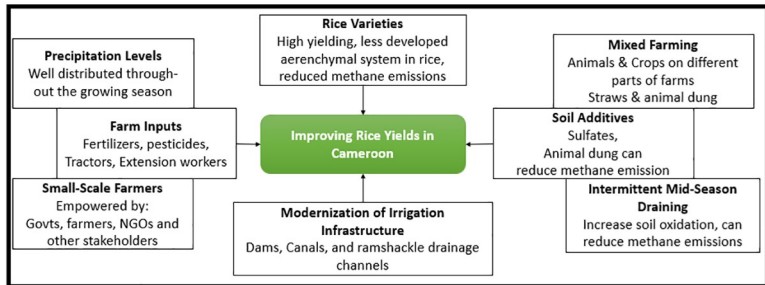

**Fig 12. Schematic of options to improve rice yields in Cameroon.** Source: Authors' conceptualization.

analysis. For example, millet with the lowest vulnerability index nationally has the highest adaptive capacity index. From a policy and farming livelihood perspective, this study suggests the modernization of Cameroon's irrigation infrastructure, use of mixed farming, farm input, soil additives, economic viability and intermittent/mid season draining to reduce the vulnerability of rice.

In the context of limitations, it should however be cautioned that, the adaptive capacity index used here is based on proxies such as literacy and poverty rates due to the difficulty associated with assessing adaptive capacity. Also, it is important to note that the time series data for the national scale analysis spans from 1961 to 2019 while that for the sub-national scale spans 1991–2016; though relatively long, more data beyond 1991 and 2016 would have better optimized the sub-national scale simulations. In addition, the absence of sufficient sub-national scale yield data repositories for the three crops involved has greatly limited the exploration at that scale. However, with the availability of time series yield data for some areas, this work has attempted to validate the vulnerability index. Finally, this study fails to compare how other approaches of simulating vulnerability simulate this data with respect to our current index. Despite these weaknesses, this study has for the first time in Cameroon provided a holistic approach by introducing the three components of vulnerability and by integrating precipitation and socio-economic variables. With this achieved, it is still important for research to be carried out on the vulnerability of other crops, verifications of future scenarios as well as crop yield gaps in Cameroon, and the introduction of more sites in the sub-national scale analysis as data becomes available. Also, it is important to note that in addition to climatic drivers, crop yields are often influenced by a complex combination of variables such as precipitation, temperatures, soils, fertilizers, high yielding varieties, crop pests and diseases as well as the socio-economic conditions of the farmers involved. Therefore, this study provides a snapshot of the role of growing season precipitation and socio-economic variables. With this achieved, it is still important for research to be carried out on the vulnerability of other crops, verifications of future scenarios as well as crop yield gaps in Cameroon. The inclusion of more sites in the sub-national scale analysis as more data becomes available should also be put into perspective. Also, it is important to note that in addition to climatic drivers, crop yields are often influenced by a complex combination of variables such as precipitation, temperatures, soils, fertilizers, high yielding varieties, crop pests and diseases as well as the socio-economic conditions of the farmers involved. Therefore, this study provides a snapshot of the role of growing season precipitation and socio-economic drivers in influencing the vulnerability of the concerned crops. Therefore, this work is proposing that adaptations to alleviate the plight of the farmers dealing with yield declines should focus on both climatic and non-climatic drivers of yield. In the context of the climatic drivers, evidence-based research and climate monitoring will provide farmers with more accurate information on changes in planting dates and future projections so that farmers can adjust and make more informed planting decisions. On the non-climatic side, interest should be placed on other pertinent drivers of crop yield such as fertilizers, high yielding varieties and crop pest and diseases. Whatever direction is adopted, it will be important for stakeholders to consider aspects of adaptation with the scope of technological developments (cloud harvesting, high yielding varieties and tractors), indigenous based aspects (knowledge of the farmers owned by virtue of their experience), economics based (interest free loans and alternative livelihoods) and social based options (help from religious and community groups, and from friends and family).

## Supporting information

**S1 Data.**
(XLSX)

## Acknowledgments

The authors would like to thank UM6P for the funding that led to the realization of this work as well as to all the authors whose works were consulted.

## Author Contributions

**Conceptualization:** Terence Epule Epule.

**Methodology:** Terence Epule Epule.

**Writing – original draft:** Terence Epule Epule, Abdelghani Chehbouni, Driss Dhiba, Daniel Etongo, Youssef Brouziyne, Changhui Peng.

**Writing – review & editing:** Terence Epule Epule, Abdelghani Chehbouni, Driss Dhiba, Daniel Etongo, Fatima Driouech, Youssef Brouziyne, Changhui Peng.

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
