## [Decision Letter · Decision Letter 0]

14 May 2021

Vulnerability of maize, millet, and rice yields to growing season precipitation and socio-economic proxies in Cameroon

PONE-D-21-13735

Dear Dr. Terence Epule Epule,

We’re pleased to inform you that your manuscript has been judged scientifically suitable for publication and will be formally accepted for publication once it meets all outstanding technical requirements.

Kind regards,

Prof. László Vasa, PhD

Academic Editor

PLOS ONE

1. Thank you for stating the following financial disclosure:

"NO -  The funders had no role in study design, data collection and analysis, decision to publish, or preparation of the manuscript."

Please provide an amended Funding Statement that declares *all* the funding or sources of support received during this specific study (whether external or internal to your organization) as detailed online in our guide for authors at http://journals.plos.org/plosone/s/submit-now. Please state what role the funders took in the study.  If any authors received a salary from any of your funders, please state which authors and which funder. If the funders had no role, please state: "The funders had no role in study design, data collection and analysis, decision to publish, or preparation of the manuscript."

Please send your amended statements by return email; we will change the online submission form on your behalf.

2. We note that Figure 1 in your submission contain map images which may be copyrighted. All PLOS content is published under the Creative Commons Attribution License (CC BY 4.0), which means that the manuscript, images, and Supporting Information files will be freely available online, and any third party is permitted to access, download, copy, distribute, and use these materials in any way, even commercially, with proper attribution. For these reasons, we cannot publish previously copyrighted maps or satellite images created using proprietary data, such as Google software (Google Maps, Street View, and Earth). For more information, see our copyright guidelines: http://journals.plos.org/plosone/s/licenses-and-copyright.

2.1. You may seek permission from the original copyright holder of Figure1 to publish the content specifically under the CC BY 4.0 license. 

2.2. If you are unable to obtain permission from the original copyright holder to publish these figures under the CC BY 4.0 license or if the copyright holder’s requirements are incompatible with the CC BY 4.0 license, please either i) remove the figure or ii) supply a replacement figure that complies with the CC BY 4.0 license. Please check copyright information on all replacement figures and update the figure caption with source information. If applicable, please specify in the figure caption text when a figure is similar but not identical to the original image and is therefore for illustrative purposes only.

Reviewers' comments:

Reviewer's Responses to Questions

**Comments to the Author**

1. Is the manuscript technically sound, and do the data support the conclusions?

Reviewer #1: Yes

Reviewer #2: Yes

2. Has the statistical analysis been performed appropriately and rigorously? 

Reviewer #1: Yes

Reviewer #2: Yes

3. Have the authors made all data underlying the findings in their manuscript fully available?

Reviewer #1: Yes

Reviewer #2: Yes

4. Is the manuscript presented in an intelligible fashion and written in standard English?

Reviewer #1: Yes

Reviewer #2: Yes

5. Review Comments to the Author

Reviewer #1: The topic of is interesting and relevant. The level of elaboration and the applied methodology is in general can be considered accurate, however a formal deficiency has to be highlighted: at the tables and figures there are no source indication which is an elementary formal requirement in case of scientific papers. They should be added, even in cases of the authors’ own-edited and calculated figures and tables.

The bibliographic review could be a broadened especially towards the experiences and lessons of other sub-Saharan African countries with the view of making the context more international. The international bibliography is rich in this respect (e.g. https://pubs.iied.org/sites/default/files/pdfs/migrate/14640IIED.pdf or https://re.volsu.ru/eng/contacts/8_Nad_i_dr.pmd.pdf .

In the conclusion part the results of the research should be developed further into concrete suggestions/proposals applicable for the further improvement of the plant sector in Cameroon (and perhaps also in other countries).

After the mentioned improvements were done, the paper can be recommended to be accepted and published.

Reviewer #2: The paper investigates the very actual problems of climate change in Cameroon, their effects on the agriculture. The research's base, the datasets and the methodology is very much appropriate. The structure of the paper is excellent, and also the content. I do think this paper will contribute to the existing knowledge.

6. PLOS authors have the option to publish the peer review history of their article (what does this mean?). If published, this will include your full peer review and any attached files.

Reviewer #1: No

Reviewer #2: No

---

## [Editor Report · Acceptance letter]

21 May 2021

PONE-D-21-13735 

Vulnerability of maize, millet, and rice yields to growing season precipitation and socio-economic proxies in Cameroon 

Dear Dr. Epule:

I'm pleased to inform you that your manuscript has been deemed suitable for publication in PLOS ONE. Congratulations! Your manuscript is now with our production department. 

Kind regards, 

on behalf of

Prof. Dr. László Vasa 

Academic Editor

PLOS ONE